# An antimicrobial drug recommender system using MALDI-TOF MS and dual-branch neural networks

Gaetan De Waele*, Gerben Menschaert, Willem Waegeman

Department of Data Analysis and Mathematical Modelling, Ghent University, Ghent, Belgium

## eLife Assessment

This **valuable** study presents a machine learning model to recommend effective antimicrobial drugs from patients' samples analyzed with mass spectrometry. The evidence supporting the claims of the authors is **convincing**. This work will be of interest to computational biologists, microbiologists, and clinicians.

*For correspondence:
gaetan.dewaele@ugent.be

Competing interest: The authors declare that no competing interests exist.

**Abstract** Timely and effective use of antimicrobial drugs can improve patient outcomes, as well as help safeguard against resistance development. Matrix-assisted laser desorption/ionization time-of-flight mass spectrometry (MALDI-TOF MS) is currently routinely used in clinical diagnostics for rapid species identification. Mining additional data from said spectra in the form of antimicrobial resistance (AMR) profiles is, therefore, highly promising. Such AMR profiles could serve as a drop-in solution for drastically improving treatment efficiency, effectiveness, and costs. This study endeavors to develop the first machine learning models capable of predicting AMR profiles for the whole repertoire of species and drugs encountered in clinical microbiology. The resulting models can be interpreted as drug recommender systems for infectious diseases. We find that our dual-branch method delivers considerably higher performance compared to previous approaches. In addition, experiments show that the models can be efficiently fine-tuned to data from other clinical laboratories. MALDI-TOF-based AMR recommender systems can, hence, greatly extend the value of MALDI-TOF MS for clinical diagnostics. All code supporting this study is distributed on PyPI and is packaged at https://github.com/gdewael/maldi-nn.

## Introduction

In diagnostic laboratories, matrix-assisted laser desorption/ionization time-of-flight mass spectrometry (MALDI-TOF MS) is routinely used for microbial species identification (*Hou et al., 2019*). Usually, microbial samples only require an overnight culturing step before being analyzed with mass spectrometry (*van Veen et al., 2010*; *Cuénod et al., 2021*). Consequently, the technology provides a time- and cost-efficient way to accurately identify the pathogen underlying an infection.

Due to the rapid evolution of antibiotic-resistant strains, it is increasingly difficult to determine a treatment based on only species identity. It has been estimated that infections caused by antibiotic-resistant bacteria have caused the deaths of 1.27 million people in 2019, making antimicrobial resistance (AMR) one of the leading causes of death on earth (*Murray et al., 2022*). Projections have estimated that this annual number could rise to 10 million by 2050 (*O'Neill, 2016*), highlighting the need for responsible antimicrobial drug use. In light of this, diagnostic laboratories will often perform various tests, such as dilution arrays or disc diffusion tests, to probe which drug will be effective (*Khan*

*et al., 2019*). Such experiments typically require further culturing and are either costly, labor-intensive, time-intensive, or a mixture of the above (*Humphries, 2022*).

Given that MALDI-TOF spectra are already routinely used for identification, it is worth investigating to which extent they can contain further information regarding the resistance status of strains (*Weis et al., 2020a*). Mining this information from the spectra could help inform healthcare workers of candidate drugs. This may nullify the need for phenotypical experiments, or (at least) direct the tests by narrowing down the choices. Furthermore, possessing a detailed resistance profile allows us to treat with more specifically working drugs (instead of broad-spectrum antibiotics) (*Weis et al., 2022*). Consequently, predicting resistance status from MALDI-TOF spectra could help toward the goals of antibiotic stewardship (*Shlaes et al., 1997*).

It has been described that some known resistance mechanisms are outside of the m/z range that MALDI-TOF spectrometers can accurately measure (*Humphries, 2022*). Still, it remains largely unknown to which extent co-evolved traits, such as subtle changes in metabolism caused by the resistance mechanism, can be detected by MALDI-TOF spectra. A number of studies have shown that some resistant strains can reliably be predicted from MALDI-TOF MS, either by identifying and detecting specific markers (e.g., peaks) or by learning patterns from data (see 'Related work'). To our knowledge, all of these studies have modeled AMR prediction for specific species–drug combinations. For this reason, they learn very specific markers of resistance, not guaranteed to extrapolate well to other drugs and species. As susceptibility rapidly evolves, it is practically impossible to perform such studies for all clinically relevant species–drug combinations. As such, the value of aforementioned studies remains of exploratory nature with limited practical value. In addition, their performance remains limited owing to small sample sizes and, likely, the inability of MALDI-TOF spectra to fully discriminate between the characteristics of interest (*Bai et al., 2017*). The recently published DRIAMS dataset (*Weis et al., 2022*) contains phenotypic AMR data covering a wide range of species and drugs, allowing to study MALDI-TOF-based AMR prediction on an unprecedented scale.

We posit that the most pertinent challenge healthcare workers face regarding AMR is to choose between all possible drugs given an infection, not whether one specific drug will be effective or not. For this reason, we argue that our models and evaluation metrics should be designed to optimally answer that question. In this study, a recommender model is proposed that can predict AMR for the whole range of pathogens and drugs encountered in clinical microbiology. In addition, species-specific recommender models for a range of common species are also trained. Our method jointly learns representations for antibiotic drugs and bacterial MALDI-TOF spectra. It can be used to recommend the most likely drug to work for any drug–spectrum combination. Consequently, the model is broadly applicable and practical to use. To summarize, our contributions are as follows:

1. We formulate a dual-branch neural network recommender system for the prediction of AMR profiles. The model operates on MALDI-TOF spectra, as well as a representation of the candidate drug.
2. We evaluate multiple state-of-the-art techniques for representing drug identity in the model.
3. We compare 'general' recommenders (trained on all spectra from all species) against species-specific recommender models
4. We perform evaluations by comparing our methods to non-recommender system baselines.
5. We show that the model efficiently transfers to data from diagnostic laboratories it was not trained on. Making the model easy to adopt for hospitals lacking the means and/or volume to collect large data.

## Related work
### MALDI-TOF-based machine learning
The most canonical task for MALDI-TOF-based methods is species identification. Identification solutions are usually provided by the MS manufacturers and are built on large, proprietary, in-house databases (*van Belkum et al., 2012*). It is unclear how these closed-source identification pipelines work, but it is likely that query spectra are directly compared to the in-house database in an approach akin to nearest neighbors (*Dauwalder et al., 2023*). While this approach works excellently for identification of most species, some strains remain problematic (*Cao et al., 2018*; *Vrioni et al., 2018*). Furthermore, by presumably focusing on the presence or absence of specific peaks, a lot of spectral information stands unused (*Florio et al., 2018*).

For various difficult prediction cases, such as strain typing, researchers often resort to machine learning (*Hettick et al., 2006*; *Wang et al., 2018*; *De Bruyne et al., 2011*). Stifled by a historical lack of large open data, machine learning research on MALDI-TOF data remains in its infancy. Most studies have narrow scopes and simple datasets (e.g., binary classification), only warranting standard preprocessing and off-the-shelf learning techniques (*Yu et al., 2022*; *Zhang et al., 2023*; *Chung et al., 2023*). Only a handful of examples exist of more advanced learning techniques specifically adapted to a MALDI-TOF-based task (*Mortier et al., 2021*; *Weis et al., 2020a*; *Vervier et al., 2015*). For a more thorough overview of MALDI-TOF-based machine learning, readers are referred to the review of *Weis et al., 2020b*.

During peer review, our attention was brought to a similar concurrent study by *Visonà et al., 2023*. Their study similarly shows that recommender systems-like models outperform more narrowly trained single-species and single-drug models. Their analysis, however, remains limited to fingerprint-based molecular representations. In addition, in this work, we demonstrate transfer learning between hospitals.

## Dual-branch neural networks

The idea of processing and combining two separate streams of information with two neural networks is applied in many fields of machine learning, collectively referred to as deep multitarget prediction (*Waegeman et al., 2019*; *Iliadis et al., 2022*).

In collaborative filtering, the goal is to predict the preference of a user to items (*He et al., 2017*). In its most elementary neural form, both users and items are represented by one-hot encodings, generating a model unable to make salient predictions for new users or items without having seen them during training. To solve this, a body of works exists on trying to communicate user- and item identity to the model via side information encoded in features (*Zheng et al., 2017*).

Dual-branch neural networks are also prevalent in language and vision. Recent advances in (multimodal) contrastive learning of image (and text) representations often rely on two neural encoders to learn a matching score between two views of the same or discordant objects (*Radford et al., 2021*; *Chen et al., 2020*). Language retrieval systems typically compare input vectors with a database of key vectors, each derived from a neural network, using approximate nearest-neighbor search techniques (*Karpukhin et al., 2020*). In biology, fields of research employing dual-branch neural networks include (1) drug–target interaction (*Lee et al., 2019*), (2) single-cell multi-omics analysis (*Lance et al., 2022*), and (3) transcription factor binding prediction (*Yang et al., 2020*), among countless others.

Most of these applications can, to varying extents, be interpreted as (collaborative filtering) recommender systems. For example, contrastive language-image models have been used to retrieve the most semantically similar images to a piece of text (*Beaumont, 2022*).

# Methods
## Data

To train models, we use the recently published DRIAMS database, consisting of 765,048 AMR measurements derived from 55,773 spectra across four different hospitals, spanning in total 74 different drugs (*Weis et al., 2022*). (These figures reflect the size of the dataset as downloaded from the original Dryad repository https://doi.org/10.5061/dryad.bzkh1899q, and after processing. For example, the number of spectra listed here corresponds to all spectra in DRIAMS for which there exists at least one AMR measurement. The total number of spectra in DRIAMS counts 250,070, but no labels are associated with these extra spectra. Further, the naming of drugs was further preprocessed such that every drug can be linked to a single chemical identifier. For more information on which drugs were merged and how this was performed, see Appendix 1.) Every drug is characterized by a canonical SMILES string obtained from PubChem (*Kim et al., 2023*). As in the original DRIAMS publication, AMR measurements are binarized according to the EUCAST norms per drug. Specifically, intermediate or resistant values are assigned a positive label, and susceptible samples a negative one. Furthermore, spectra are identically processed as in the original publication. Briefly, the following steps are performed: (1) square-root transformation of the intensities, (2) smoothing using a Savitzky–Golay filter with half-window size of 10, (3) baseline correction using 20 iterations of the SNIP algorithm, (4) trimming to the 2000–20,000 Da range, (5) intensity calibration so that the total intensity sums to 1, and (6) binning

the intensities by summing all values in intervals of 3 Da. After preprocessing, every spectrum is represented as a 6000-dimensional vector.

The main experiments concern models that are trained on data from one hospital only (DRIAMS-A, University Hospital Basel). All spectra and measurements derived from the other three hospitals in DRIAMS are left out for transfer learning experiments (see Results). Within DRIAMS-A, all spectra from before 2018 are allocated to the training set, and all spectra measured during 2018 are evenly split between validation and test set. This split in time reflects a realistic evaluation scenario, as models trained on historical data need to generalize to new patients possibly infected by newly evolved strains. The final sizes of all splits are as follows: 409,395 labels across 28,331 spectra for the training set, 76,431 labels across 4994 spectra for the validation set, and 76,133 labels across 4999 spectra for the test set.

## Metrics

The main objective of this study is to train models to effectively recommend treatments for patients. Hence, unless otherwise noted, metrics are computed on a per-patient basis, and then averaged. This is equivalent to macro-averaged metrics, but then computed per instance (spectrum), instead of per class (drug) (*Waegeman et al., 2019*). For simplicity, we omit the 'macro' prefix from metrics, and – unless otherwise indicated – always use spectrum-macro metrics.

The area under the receiver operating characteristic curve (ROC-AUC) measures the probability that any positive (resistant or intermediate) sample is assigned a higher predicted probability of being positive as compared to any negative (susceptible) sample. It is a measure of the average quality of the ranking of suggested drugs to a patient. To compute the (per-patient average) ROC-AUC, for any spectrum/patient, all observed drug resistance labels and their corresponding predictions are gathered. Then, the patient-specific ROC-AUC is computed on that subset of labels and predictions. Finally, all ROC-AUCs per patient are averaged to a 'spectrum-macro' ROC-AUC.

The Precision at 1 of the negative class (Prec@1(-)) evaluates how often the top-ranked prediction is correct. Hence, in this case, it reports the proportion of cases for which the 'most likely susceptible

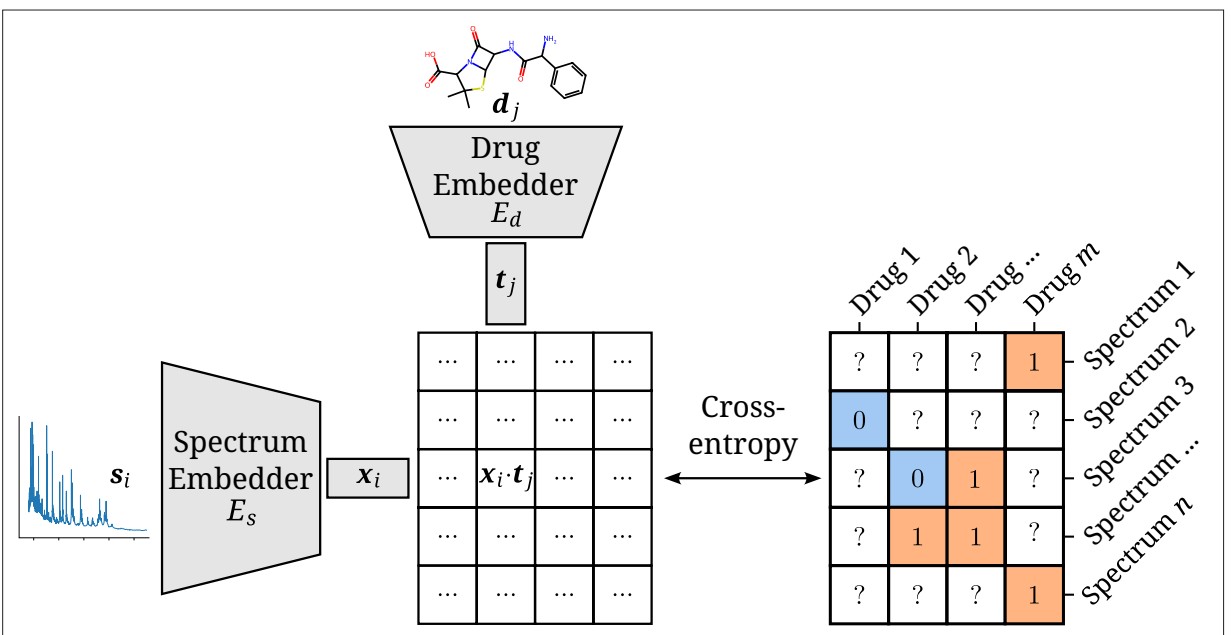

**Figure 1.** Architectural overview of the proposed model. Antimicrobial resistance (AMR) labels of spectrum–drug pairs can be represented in an incomplete matrix. A microbial sample that is susceptible to a drug is denoted by a negative label (orange), whereas positive labels (blue) signify an intermediate or resistant combination. Instance (spectrum) and target (drug) embeddings $x_i$ and $t_j$ are obtained from their respective input representations passed through their respective neural network branch. The two resulting embeddings are aggregated to a single score by their (scaled) dot product. The cross-entropy loss optimizes this score to be maximal or minimal for positive or negative combinations of microbial spectra and drugs, respectively.

**Table 1.** All tested model sizes for the (instance) spectrum branch.
Hidden sizes represent the evolution of the hidden state dimensionality as it goes through the model, with every hyphen defining one fully connected layer. The listed number of parameters only includes those of the instance (spectrum) branch.

| Size | # weights | Hidden sizes |
|------|-----------|--------------|
| S | 1,578,176 | 6000-256-128-64 |
| M | 3,246,784 | 6000-512-256-128-64 |
| L | 6,846,144 | 6000-1024-512-256-128-64 |
| XL | 15,093,440 | 6000-2048-1024-512-256-128-64 |

drug' prediction is actually an effective one. In a scenario where the top recommended drug is always administered, it corresponds to the percentage of correctly suggested treatments.

## Model architecture

We formulate AMR prediction as a multitarget classification problem with side information for both instances and targets, also referred to as dyadic prediction (*Waegeman et al., 2019*). In this context, let us denote a sample in the dataset $\mathcal{D}$ by a triplet $(s_i, d_j, y_{ij})$, where $y_{ij}$ denotes the resistance label of a microbial spectrum $s_{i \in \{1,...,n\}}$ w.r.t. a drug $d_{j \in \{1,...,m\}}$. This dataset can be arranged in an incomplete score matrix $Y \in \{0,1\}^{n \times m}$. In what follows, the final architectural set-ups used to present the results are described. For details on hyperparameter tuning, readers are referred to Appendix 2.

The model consists of two separate neural network embedders $E_s(\cdot)$ and $E_d(\cdot)$ for processing the spectra and drugs, respectively. The resulting instance and target embeddings $x_i$ and $t_j$ are then combined into a single score by their scaled dot product $\hat{y} = \frac{x_i \cdot t_j}{\sqrt{h}}$ (*Rendle et al., 2020*). The scaling factor $\sqrt{h}$, with $h$ the dimensionality of both embeddings, is inspired by the formulation of self-attention (*Vaswani et al., 2017*). It ensures the dot products to be of manageable magnitudes, even for large values of $h$. This score can be used together with the sigmoid function and the cross-entropy loss to optimize the two-branch neural network to map a spectrum–drug pair to a resistance label (*Iliadis et al., 2022*). An overview of the model is visualized in *Figure 1*.

The representations of the instance vectors $x_i$ are extracted from a neural network $E_s(\cdot)$ operating on the processed and binned MALDI-TOF spectra $s_i$. $E_s(\cdot)$ is parameterized by a multi-layer percep-tron (MLP), consisting of a series of fully connected layers. Between every two such layers, a series of operations consisting of (1) a GeLU activation (*Hendrycks and Gimpel, 2016*), (2) a dropout rate of 0.2 (*Srivastava et al., 2014*), and (3) layer normalization (*Ba et al., 2016*) is applied. We include multiple model sizes in our final results (*Table 1*). To make comparisons easier, all models output the same number of hidden dimensions that are used in the dot product, $x_i \in \mathbb{R}^{64}$.

Drug identity can be communicated to the model in a number of ways. In this work, we study the following different input representations $d_j$ and embedder $E_d(\cdot)$ combinations:

1. As indices in a one-hot encoding paired with a single linear layer.
2. As Extended Connectivity Fingerprints paired with a single linear layer.
3. As DeepSMILES strings (*O'Boyle and Dalke, 2018*) paired with a 1D convolutional neural network (CNN).
4. As DeepSMILES strings paired with a gated recurrent unit neural network (GRU).
5. As DeepSMILES strings paired with a transformer neural network.
6. As images paired with a 2D CNN.
7. As rows of a pre-computed string kernel on the SMILES strings (LINGO *Vidal et al., 2005*), paired with a single linear layer.

For all these combinations, the embedder outputs target embeddings $t_j \in \mathbb{R}^{64}$. For more details on the different drug embedders and their hyperparameters (as well as their tuning), see Appendix 2. For every combination of spectrum embedder (four sizes: S, M, L, and XL) and drug embedder (seven types), six different learning rates ({1e-5, 5e-5, 1e-4, 5e-4, 1e-3, 5e-3}) are tested. For all these different combinations, five models are trained (using different random seeds for model initialization and batching of data). For every spectrum and drug embedder combination, only results from the

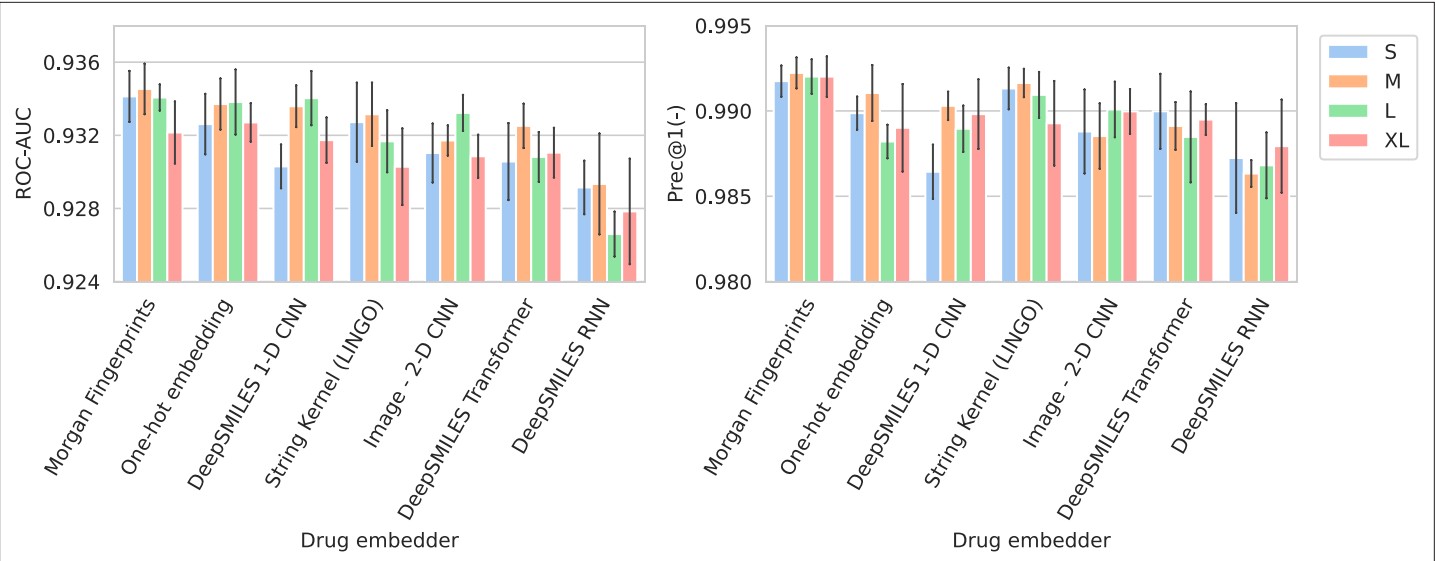

**Figure 2.** Barplots showing test performance results for all trained models. Area under the receiver operating characteristic curve (ROC-AUC) evaluates overall ranking of predictions. Prec@1(-) evaluates how often the top suggested treatment would be effective. Both metrics are calculated per spectrum/patient and then averaged. Errorbars represent the standard deviation over five random model seeds. The x-axis and colors show the different drug and spectrum embedders, respectively.

best learning rate are presented; that is, the learning rate resulting in the best average validation ROC-AUC for that combination.

All models are trained with the Adam optimizer (*Kingma and Ba, 2014*) for a maximum of 50 epochs with a batch size of 128. A linear learning rate warm-up over the first 250 steps is applied, after which the rate is kept constant. As every epoch constitutes one pass over every label and, hence, multiple passes over every individual drug and spectrum, a branch can technically already be overfitting before the end of the first epoch. Because of this, performance on the validation set is checked every tenth of an epoch. Training is halted early when validation ROC-AUC has not improved for 10 validation set checks. The checkpoint of the best performing model (in terms of validation ROC-AUC) is used as the final model.

## Results

The following section will first relay the results of the different dual-branch model configurations. Afterward, the 'general' AMR recommender is matched up against 'species-specific' and 'species–drug-specific' models. Finally, the models' capabilities and representations are examined through transfer learning and embeddings.

### Encoding species and drugs effectively

*Figure 2* shows the performance of all trained models in terms of their average ROC-AUC and Prec@1(-). It can be seen that, in general, performance differences between model configurations occupy a small margin. However, trends can still be found. Models using Morgan fingerprints typically outperform other drug embedding strategies. Morgan fingerprints provide a compressed and preprocessed input format, the nature of which provides an apparent advantage over input representations that require more pattern extraction. The small number of different antimicrobial drugs may not be conducive to learning complex representations. Indeed, embedding drugs without a compound information (i.e., one-hot embedding) is a competitive approach for this problem, resulting in the – on average – second best models in terms of ROC-AUC. On the spectrum embedder side, it is observed that the medium or large variants typically perform best. The full ROC curve (showing sensitivity and specificity) for the best-performing model is shown in *Appendix 3—figure 1*.

Performance in terms of Macro ROC-AUC can be found in *Appendix 3—figure 2*. The Macro ROC-AUC averages the ROC-AUC for every individual drug. Here, Morgan fingerprints similarly reach the best performances. The full list of performances can be found in *Appendix 3—table 1*.

In *Appendix 3—figure 3*, the performance of the spectrum embedder sizes is compared against a linear baseline. The linear baseline uses the same preprocessed input spectrum representation, but only uses a single linear combination to produce an embedding. For this comparison, only the Morgan fingerprint drug embedders are used as they produce the best-performing models overall. Models using nonlinear multi-layer spectrum embedders obtain considerably better performance over linear embedders.

## Species-specific models improve recommendation

The recommender systems presented in the previous section provide an incredibly general tool. Trained as single models for all species and drugs, their versatility is unparalleled compared to previous studies that create classifiers for specific drug–species combinations (*Weis et al., 2020b*). In between the extremes of 'one model for everything' and 'a model per species and per drug', there lies a compromising approach: a species-specific recommender system for all drugs. Such recommender systems would be more specialized in nature, but their usefulness hinges upon having done prior species identification. As these are typically included in the MS' manufacturer's software, a more specialized species-specific recommender may provide better performance without incurring extra cost. The disadvantage of such models is that (1) they cannot be used for species for which there is not enough data to train a separate model (i.e., rarely occurring species), and (2) they rely on the prior identification step to be correct.

Here, we create species-specific recommender models for the 25 most occurring species in DRIAMS-A. The training setup for these models is kept the same as in the previous section. The difference between 'general' recommenders and 'species-specific recommenders' is that each species-specific recommender model is only trained on the subset of data covering their respective species (as these models use a smaller training set, validation is checked every fourth of an epoch instead of every tenth). Together, the test predictions of the 25 species-specific recommenders cover 4229 spectra, 56 drugs, and 69,827 AMR labels (covering 91.27% of the original test set). *Table 2* compares the two best 'general' recommenders from the previous section to their species-specific recommender counterparts. It is observed that species-specific recommenders deliver better predictions across all evaluated metrics.

As opposed to the species-specific models, the 'general' recommender can use learned representations from one species to enhance predictions for other species, benefitting from multitask learning. The fact that this latter mode of learning performs worse on this problem, however, indicates that such transfer of learned knowledge is of limited usefulness for AMR prediction. Still, the 'general' recommender model remains useful in instances where the species could not be identified, or is rare. In *Appendix 3—table 2*, the 25 species for which specific recommenders were trained are listed, along with their performances.

**Table 2.** Test performance of selected general and species-specific dual branch recommender models.

The listed averages and standard deviations are calculated over five independent runs of the same model. Performance is computed on the subset of labels spanning the 25 most common species in DRIAMS-A.

| Model | ROC-AUC | Prec@1(-) | Macro ROC-AUC |
|---|---|---|---|
| General recommender (Morgan fingerprints – M) | 0.9411 ± 0.0007 | 0.9967 ± 0.0011 | 0.7684 ± 0.0050 |
| General recommender (one-hot – L) | 0.9408 ± 0.0011 | 0.9940 ± 0.0009 | 0.7746 ± 0.0316 |
| Species-specific recommenders (Morgan fingerprints – M) | 0.9461 ± 0.0010 | **0.9973 ± 0.0004** | **0.7905 ± 0.0151** |
| Species-specific recommenders (one-hot – L) | **0.9468 ± 0.0012** | 0.9950 ± 0.0011 | 0.7686 ± 0.0155 |

ROC-AUC, area under the receiver operating characteristic curve.

## Dual-branch recommenders improve over baselines

In order to gain better insight into the performance of our models, in this section, both our 'general' and 'species-specific' recommenders are squared up against extensive baselines.

Previous studies have studied AMR prediction in specific species–drug combinations. For this reason, it is useful to compare how the dual-branch setup weighs up against training separate models for separate species and drugs. In *Weis et al., 2020b*, for example, binary AMR classifiers are trained for the following three combinations: (1) *Escherichia coli* with ceftriaxone, (2) *Klebsiella pneumoniae* with ceftriaxone, and (3) *Staphylococcus aureus* with oxacillin. Here, such 'species–drug-specific classifiers' are trained for the 200 most common combinations of species and drugs in the training dataset. For these combinations, binary logistic regression, XGBoost (*Chen and Guestrin, 2016*), and MLPs are tested. The tested MLPs come in the same four sizes as the spectrum branches of the dual-branch models. Other than having an output node of size 1 for binary classification, they share all hyperparameters with the tested spectrum branches. For details on the training and tuning procedure of all baselines, see Appendix 2.

There exist many species–drug combinations for which there are either only positive or only negative labels. As it is impossible to train and evaluate models for these cases, models are trained only for the 200 most occurring combinations for which both labels are present in the training, validation, and test set. We refer to these models as 'species–drug classifiers'.

In addition, it is useful to probe model performance against what experts may be able to guess. Given knowledge of the species identity in question, an expert will – in many cases – already be able to make a good guess toward what drugs will be effective or not. Hence, baseline 'best guess' performance would not result in a ROC-AUC of 0.5. A way to simulate such 'expert's best guess' baseline predictions is through counting label frequencies in the training set. More specifically, for a test label belonging to a certain species and drug, the labels in the training set corresponding to that drug and species can be gathered. The frequency by which that training set is positive or negative can be used to infer a test predicted probability. We refer to this baseline as 'simulated expert's best guess'. More formally, considering all training spectra as $\mathcal{S}_{\text{train}}$, all training labels corresponding to one drug $j$ and

**Table 3.** Test performance of selected recommender models, compared to the performance of a collection of models – each trained on only one species–drug combination – coined 'species–drug classifiers'.

'Species–drug classifiers' refer to a collection of binary classifiers, each trained to predict antimicrobial resistance (AMR) status for a subset of data comprising a single species–drug combination. 'Simulated expert's best guess' refers to counting AMR label frequencies in single species–drug combinations and taking those as predictions. The listed averages and standard deviations are calculated over five independent runs of the same model. Given the non-stochastic nature of the logistic regression and XGBoost implementations, only one set of models is trained and, hence, no standard deviations are reported. Performance is computed on the subset of labels spanning the 200 most common species–drug combinations.

| Model | ROC-AUC | Prec@1(-) | Macro ROC-AUC | Species–drug macro ROC-AUC |
|---|---|---|---|---|
| Species-specific recommenders (Morgan fingerprints – M) | 0.9009 ± 0.0018 | **0.9830 ± 0.0015** | **0.8283 ± 0.0059** | 0.6381 ± 0.0121 |
| Species-specific recommenders (one-hot – L) | **0.9030 ± 0.0018** | 0.9814 ± 0.0020 | 0.8129 ± 0.0079 | 0.6511 ± 0.0290 |
| General recommender (Morgan fingerprints – M) | 0.8939 ± 0.0016 | 0.9746 ± 0.0006 | 0.8114 ± 0.0064 | 0.6517 ± 0.0076 |
| General recommender (one-hot – L) | 0.8933 ± 0.0020 | 0.9778 ± 0.0023 | 0.8124 ± 0.0033 | 0.6521 ± 0.0078 |
| Species–drug classifiers (MLP – S) | 0.8341 ± 0.0135 | 0.9420 ± 0.0123 | 0.8005 ± 0.0032 | 0.6745 ± 0.0218 |
| Species–drug classifiers (MLP – M) | 0.8382 ± 0.0077 | 0.9421 ± 0.0196 | 0.8075 ± 0.0049 | 0.6797 ± 0.0097 |
| Species–drug classifiers (MLP – L) | 0.8457 ± 0.0088 | 0.9505 ± 0.0100 | 0.8037 ± 0.0079 | 0.6648 ± 0.0149 |
| Species–drug classifiers (MLP – XL) | 0.8611 ± 0.0049 | 0.9722 ± 0.0041 | 0.8106 ± 0.0069 | 0.6801 ± 0.0101 |
| Species–drug classifiers (logistic regression) | 0.8684 | 0.9432 | 0.7989 | **0.7200** |
| Species–drug classifiers (XGBoost) | 0.8346 | 0.9196 | 0.7763 | 0.6236 |
| Simulated expert's best guess | 0.8681 | 0.9743 | 0.7159 | 0.5000 |

ROC-AUC, area under the receiver operating characteristic curve.

species $t$ are gathered: $\mathcal{Y}^{j,t}_{\text{subset}} = \{y_{ij} \mid s_i \in \mathcal{S}_{\text{train}} \land \text{species}(s_i) = t\}$. The 'simulated expert's best guess' predicted probability for any spectrum $s_i$ and drug $d_j$, then, corresponds to the fraction of positive labels in their corresponding training label set $\mathcal{Y}^{j,t}_{\text{subset}}$: $\Pr\left(y_{ij} = 1 \mid \text{species}(s_i) = t, d_j\right) = \frac{\sum_{y \in \mathcal{Y}^{j,t}_{\text{subset}}} y}{|\mathcal{Y}^{j,t}_{\text{subset}}|}$.

*Table 3* compares the recommenders from the previous section to non-recommender baselines. As the baselines are only trained on the 200 most common species–drug combinations, performance is computed on that subset of test labels. This reduced test set spans 4017 spectra, 35 drugs, and 53,503 labels (covering 70.28% of the original test set). Dual-branch recommenders outperform baselines on all but one metric. Logistic regression baselines result in the best average ROC-AUC for individual species–drug combinations. By all other metrics, dual-branch recommenders outshine a collection of species–drug-specific classifiers. It is illustrated that, when the question is to choose between drugs for a patient (evaluated by the patient-averaged ROC-AUC or Prec@1(-)), a model designed as a recommender will outperform binary classification models trained to predict AMR for specific drugs. On the other hand, species-specific binary classifiers are optimal for distinguishing spectra for a specific drug. The crux of our case in favor of recommender models relies, hence, on the fact that patient-averaged metrics are more representative of AMR models' utility in clinical diagnostics.

It is useful to note that *any* gain in performance over the 'simulated expert' means that AMR signal could be mined from the spectra. Hence, any performance above this level results in a real-world information gain for clinical diagnostic laboratories.

## Efficient transfer learning to new hospitals

An AMR prediction model trained using data from one hospital may not be suitable for use in other hospitals for several reasons. First, protocols such as sample preparation and culturing media differ from hospital to hospital, resulting in systematic differences in MALDI-TOF spectra (*Weis et al., 2022*). Second, epidemiology is spatially varied. Drug-resistant clades may be prevalent in one region or country, but absent in another (*Humphries, 2022*). Finally, the MALDI-TOF instruments themselves may also be specific to the hospital and influence the readout. This influences prediction models, as a hospital-specific effect is reported by the study introducing the DRIAMS dataset (*Weis et al., 2022*). They find that models typically perform best when trained with data from the same hospital. Here, hospital transferability is studied in the context of transfer learning.

Data from DRIAMS-B, -C, and -D are split into training, validation, and test set. The train set for these hospitals consists of 1000 randomly drawn spectra, simulating a small data scenario where the hospital has not spent considerable efforts in data collection. The remaining spectra for all three hospitals are evenly split among validation and test set.

For all three hospitals, we train models in the same way as previously (see Methods). A comparison is made between fine-tuning starting from models trained on DRIAMS-A (i.e., models from previous sections) and dual-branch models trained from scratch (*Figure 3*). For simplicity, we transfer the non-species-specific, 'general' recommenders as we feel this reflects a more realistic use case for labs that cannot afford to gather spectra for all possible species and additionally fine-tune them. Over all three hospitals, models fine-tuned from a DRIAMS-A checkpoint generally outperform models trained from scratch. This trend holds true over different numbers of spectra available in the fine-tuning set. In

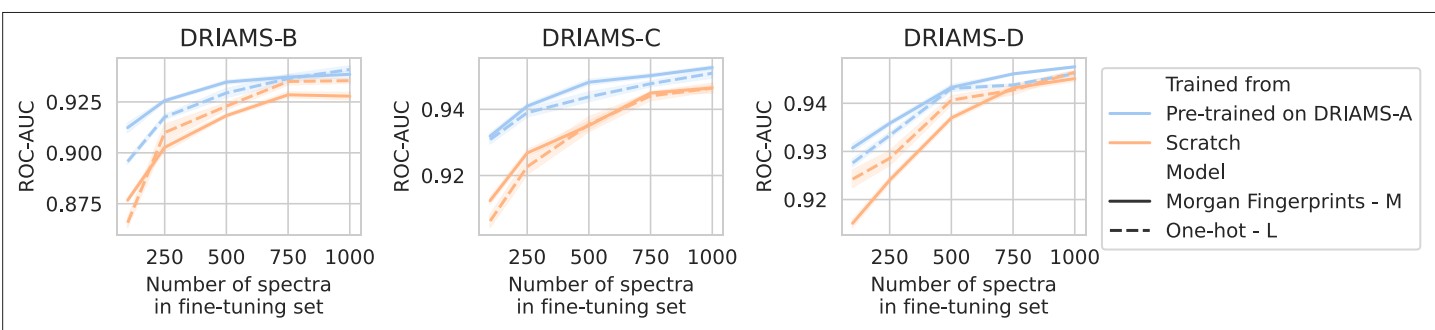

**Figure 3.** Transfer learning of DRIAMS-A models to other hospitals. Errorbands show the standard deviation over five runs. Results in terms of other evaluation metrics are shown in *Appendix 3—figure 4*.

general, it can be seen that pre-trained models require very little fine-tuning spectra to obtain performances in the same order of magnitude as with DRIAMS-A (see previous Results sections). Performance comparisons of the same models in terms of other metrics are shown in *Appendix 3—figure 4*.

Lowering the amount of data required is paramount to expedite the uptake of AMR models in clinical diagnostics. The transfer learning qualities of dual-branch models may be ascribed to multiple properties. First of all, since different hospitals use much of the same drugs, transferred drug embedders allow for expressively representing drugs out of the box. Secondly, owing to multitask learning, even with a limited number of spectra, a considerable fine-tuning dataset may be obtained, as all available data is 'thrown on one pile'.

## MALDI-TOF spectra embeddings

To investigate what the dual-branch models have learned to represent, MALDI-TOF spectra embeddings are examined. For this purpose, both the best-performing 'general' recommender and

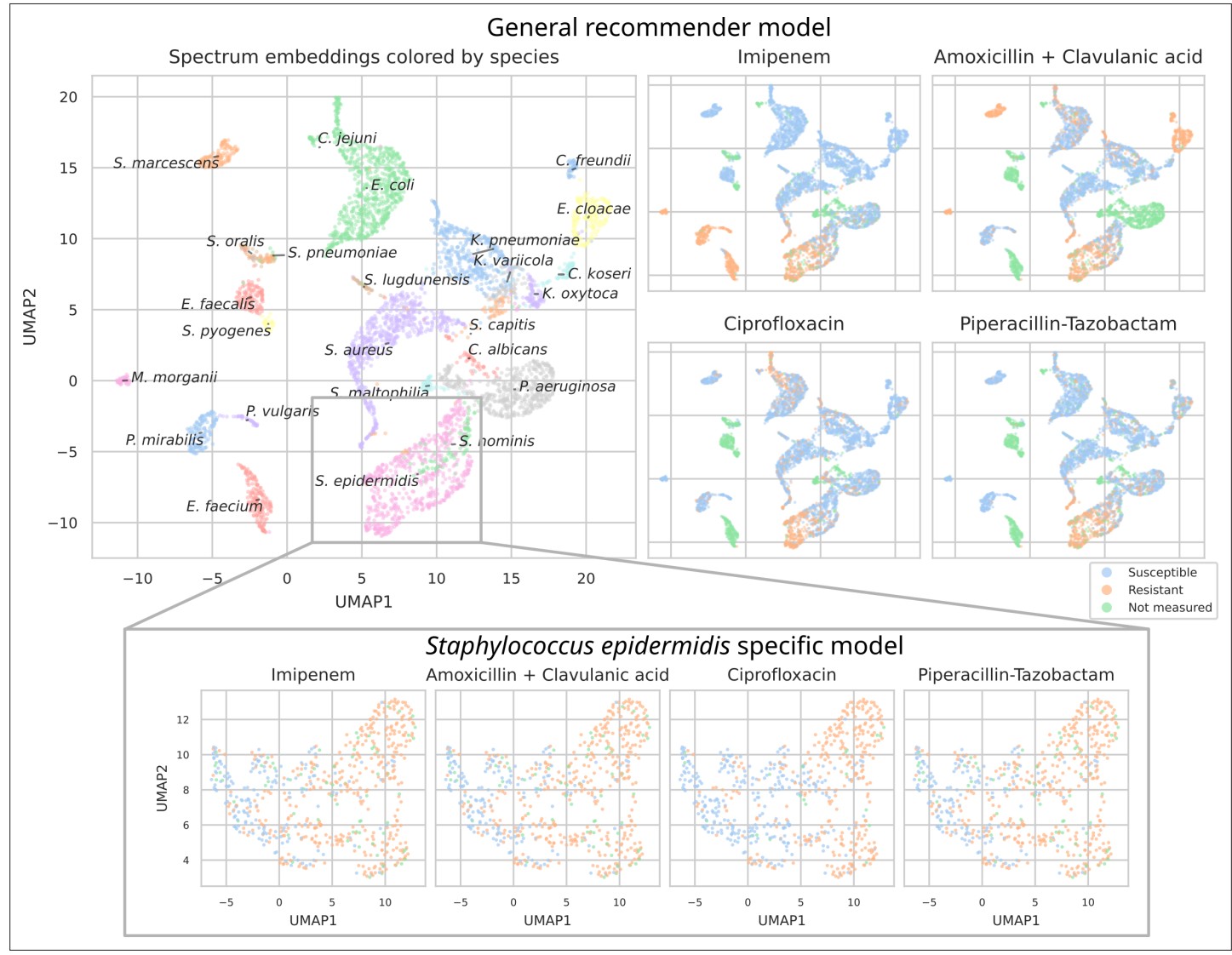

**Figure 4.** UMAP scatterplots of test set matrix-assisted laser desorption/ionization time-of-flight (MALDI-TOF) spectra embeddings $x_i$.
Top: embeddings from a 'general' (trained on all species) recommender. Only embeddings belonging to the 25 most occurring species in the test set are shown. The panels on the right show the same embeddings as on the left, but colored according to its antimicrobial resistance (AMR) status to a certain drug. The four displayed drugs are selected based on a ranking of the product of the number of positive and negative labels $\sum_{i=1}^{n}[y_{ij} = 0] \cdot \sum_{i=1}^{n}[y_{ij} = 1]$. In this way, the drugs that have a lot of observed labels, both positives and negatives, are displayed. Bottom: highlighted embeddings from a *S. epidermidis*-specific recommender model.

'species-specific' recommender are used. Here, we visualize the embeddings $x_i \in \mathbb{R}^{64}$ of all test set spectra from the 25 most occurring pathogens. To visualize in a two-dimensional space, UMAP is applied (using default parameters apart from min_dist = 0.5) (Increasing this parameter helps reduce UMAP packing points too tightly together, hence, making for a more-legible plot.). *Figure 4* shows the resulting embeddings, colored by species identity, as well as by their AMR status to a selection of drugs.

The MALDI-TOF embeddings from the 'general' recommender model are grouped primarily per species. This shows that, without being instructed to discriminate between species, the model has learned to group spectra of the same species together. Furthermore, species under the same genus are typically grouped close together, illustrating that the model can pick up hierarchical relations in the tree of life from the data. Within species clusters, the AMR status subplots show that samples are often grouped according to their resistance. For example, for *Staphylococcus epidermidis* and *S. aureus*, multidrug-resistant variants clearly form subclusters. In addition, the cluster of *E. coli* spectra shows a clear tail with samples resistant to ciprofloxacin. Embeddings from the species-specific recommender models show this phenomenon more clearly. UMAP embedding plots from the 'general recommender' colored by other drugs are shown in *Appendix 3—figure 5*. In addition, species-specific recommender system embeddings for some prominent species are shown in *Appendix 3—figure 6*.

## Discussion

Prior work on AMR prediction has always modeled within the boundaries of one clade and drug (class), using standard machine learning practices. This work differentiates itself from others by constructing one model for the whole range of drugs encountered in clinical diagnostics. We propose to model AMR prediction via dual-branch neural networks, producing a novel MALDI-TOF-based AMR recommender system. The proposed models come with improved performance over the approaches taken in previous works.

In clinical diagnostics, AMR predictions could be used to decide which drug to administer on a per-patient basis. For this reason, we argue that evaluation metrics should probe the average quality of predictions per patient (i.e., spectrum-macro metrics). We show that, for these metrics, recommender systems consistently outperform baselines.

We postulate that the performance of the proposed models is still limited due to (1) lacking a MALDI-TOF-specific learning architecture, (2) collection of more data, especially on rarely encountered species and drugs, and (3) inherent technological limitations of MALDI-TOF MS. Whilst the former is the subject of further machine learning research, the latter two can be considered by equipping the model with some notion of uncertainty, epistemic and aleatoric, respectively (*Hüllermeier and Waegeman, 2021*). In medical decision-making applications, effective uncertainty estimates would be an invaluable tool to aid understanding the models' predictions. A fourth factor to consider is that perfect test set performance may also be unattainable due to labeling errors. This comes as a consequence of (1) error-prone laboratory measurements of minimum inhibitory concentration (MIC) values, and (2) the fact that EUCAST norms change over time, resulting in outdated label thresholds for historical data.

As bacterial strains readily adapt resistance to new and frequently used antibiotics, it is impossible for an AMR model to maintain its performance over time. Consequently, an obvious need for continual data collection and online machine learning approaches presents itself. It is for this reason that ML for AMR prediction will prove most valuable when integrated tightly in the inner workings of healthcare (*Lee and Lee, 2020*).

It stands to reason that blindly following the recommender system's predictions spells misery. For example, healthcare practitioners should additionally take into account host-specific factors such as patient age, medical history, and concurrent medication. Additionally, as the model is trained on the whole repertoire of antimicrobial drugs, it will have learnt that broad-spectrum antibiotics are typically effective. Hence, it may overrecommend their use. As a consequence, the model's proposed treatment strategies may not be aligned with antibiotic stewardship, instead exacerbating the very issue it is designed to mitigate. To tackle this problem, one could downweigh the prediction probabilities of undesirable drugs, or, alternatively, train a dual-branch model on only more specifically working drugs.

In summary, this study serves as the first proof of concept for large MALDI-TOF-based antimicrobial drug recommenders. In this context, we highlight the need for appropriate metrics, proposing that per-patient metrics are most suitable. Extensive experiments on our proposed dual-branch model allow us to assemble some conclusions w.r.t. its use. Firstly, we find that medium-sized MLP spectrum embedders (counting 3.2 M weights) generally perform best. Second, incorporating chemical information works best using Morgan fingerprints. Third, while more data may skew the favor toward the other side, given the current available data, species-specific models outperform recommenders trained for all species. For the smaller datasets used in the transfer learning experiments, the structural inductive bias lent to the model via Morgan fingerprints delivers best results. Our experiments demonstrate that dual-branch recommenders outperform non-recommender baselines on relevant metrics. In the above discussion, some considerations are listed w.r.t. its practical implementation in healthcare. Taken together, this work demonstrates the potential of AMR recommenders to greatly extend the value of MALDI-TOF MS for clinical diagnostics.

## Acknowledgements

This work was supported by the Research Foundation – Flanders (FWO) (PhD fellowship fundamental research grant 1153024N to GDW). WW also received funding from the Flemish Government under the 'Onderzoeksprogramma Artificiële Intelligentie (AI) Vlaanderen' Programme.

## Additional information

### Funding

| Funder | Grant reference number | Author |
|---|---|---|
| Fonds Wetenschappelijk Onderzoek | 1153024N | Gaetan De Waele |
| Vlaamse regering | Onderzoeksprogramma Artificiële Intelligentie (AI) Vlaanderen | Willem Waegeman |

The funders had no role in study design, data collection and interpretation, or the decision to submit the work for publication.

### Author contributions

Gaetan De Waele, Conceptualization, Formal analysis, Investigation, Visualization, Methodology, Writing - original draft; Gerben Menschaert, Conceptualization, Supervision, Project administration, Writing - review and editing; Willem Waegeman, Conceptualization, Supervision, Funding acquisition, Project administration, Writing - review and editing

### Author ORCIDs

Gaetan De Waele (iD) https://orcid.org/0000-0003-0367-9699

Joint public reviews: https://doi.org/10.7554/eLife.93242.4.sa1
Author response https://doi.org/10.7554/eLife.93242.4.sa2

## Additional files

### Supplementary files

• MDAR checklist

### Data availability

The current manuscript is a computational study, so no data have been generated for this manuscript. All experiments build upon publicly available data and code, all instructions to reproduce our study are available at https://github.com/gdewael/maldi-nn (copy archived at *De Waele, 2024*).

The following previously published dataset was used:

| Author(s) | Year | Dataset title | Dataset URL | Database and Identifier |
|---|---|---|---|---|
| Weis C, Cuenod A, Rieck B, Borgwardt K, Egli A | 2021 | DRIAMS: Database of Resistance Information on Antimicrobials and MALDI-TOF Mass Spectra | https://doi.org/10.5061/dryad.bzkh1899q | Dryad Digital Repository, 10.5061/dryad.bzkh1899q |

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

# Appendix 1

## DRIAMS processing

As our models require every target to correspond to one specific drug (for which a SMILES string can be obtained), data provided by *Weis et al., 2022* is further cleaned up. First, as 'quinolones' and 'aminoglycosides' constitute classes of drugs rather than single ones, these drugs and their corresponding measurements are removed from the dataset. Second, some drug names in DRIAMS that refer to the same chemical structure are merged to a single drug. As this merging of drugs also combines their labels, care is taken so that no conflicting labels are combined. If, for a single spectrum, labels exist for both of the merging drugs in question, the label is only kept if both measurements are congruent (either both resistant, intermediate, or susceptible). Otherwise, the merged label is discarded. Finally, some drugs are renamed such that there is less ambiguity as to exactly which compound is referred to by their name. The full list of modifications to drug names is provided in *Appendix 1—table 1*.

**Appendix 1—table 1.** Full list of modifications made to drug names in DRIAMS.
Modifications consist of (1) removal of drugs, (2) merging of drugs, and (3) renaming drugs.

| Original drug name | Step undertaken |
| --- | --- |
| Quinolones | Removed |
| Aminoglycosides | Removed |
| Ofloxacin | Merged with levofloxacin |
| Benzylpenicillin | Merged with penicillin |
| Benzylpenicillin_others | Merged with penicillin |
| Benzylpenicillin_with_meningitis | Merged with penicillin |
| Benzylpenicillin_with_pneumonia | Merged with penicillin |
| Penicillin_with_endokarditis | Merged with penicillin |
| Penicillin_without_endokarditis | Merged with penicillin |
| Penicillin_without_meningitis | Merged with penicillin |
| Penicillin_with_meningitis | Merged with penicillin |
| Penicillin_with_pneumonia | Merged with penicillin |
| Penicillin_with_other_infections | Merged with penicillin |
| Cefuroxime.1 | Merged with cefuroxime |
| Cotrimoxazol | Merged with cotrimoxazole |
| Gentamicin_high_level | Merged with gentamicin |
| Cefoxitin_screen | Merged with cefoxitin |
| Teicoplanin_GRD | Merged with teicoplanin |
| Vancomycin_GRD | Merged with vancomycin |
| Rifampicin_1mg-l | Merged with rifampicin |
| Meropenem_with_meningitis | Merged with meropenem |
| Meropenem_without_meningitis | Merged with meropenem |
| Meropenem_with_pneumonia | Merged with meropenem |
| Amoxicillin-Clavulanic acid_uncomplicated_HWI | Merged with amoxicillin-clavulanic acid |
| Strepomycin_high_level | Renamed to streptomycin |
| Bacitracin | Renamed to bacitracin A |
| Ceftarolin | Renamed to ceftaroline fosamil |

*Appendix 1—table 1 Continued on next page*

*Appendix 1—table 1 Continued*

| Original drug name | Step undertaken |
| --- | --- |
| Fosfomycin-Trometamol | Renamed to fosfomycin tromethamine |

To present drugs to the model, all names of drugs are converted to SMILES strings. In this work, PubChem's canonical SMILES strings of every compound are used. In PubChem, canonical SMILES are not isomeric, which means that stereochemistry is ignored. As such, two drugs that are stereoisomers are treated as a single drug, this is the case for ofloxacin and levofloxacin. Furthermore, many drugs in the dataset refer to the co-administration of two compounds (such as, for example, ampicillin-sulbactam or amoxicillin-clavulanic acid). These cases are treated as a single drug with a SMILES string consisting of the strings of both constituent compounds separated by a '.' character, as is common practice with SMILES strings.

## Appendix 2

## Modeling set-up

### Drug embedders

In this article, seven ways to encode drugs in a model are tested out. In this section, those seven drug embedders are described in detail. All descriptions correspond to the final set-up used to present results, hyperparameter tuning results are presented in Appendix 2.

All drug embedders encode drugs to a vector $t_j \in \mathbb{R}^{64}$. The most simple way to obtain a dense vector of that size for every drug is via a **one-hot embedding**. Every drug gets assigned an index in a vector, and the resulting vectors are embedded to a dense representation via a single linear layer. Encoding drugs in this way is the most straightforward, but no structural information of the underlying active compound is included. No inductive bias is presented to the model that will give structurally similar drugs comparable embeddings. As such, all this information must be learnt from data. Similarly, such drug embedders cannot be generalized out-of-the-box to drugs it has not seen in the training data, as there are no indices – and learnt embeddings – for them.

The (local) structure of drugs can be encoded via fingerprints. A molecular fingerprint corresponds to a bit-vector in which every bit corresponds to the presence or absence of a substructure (*Capecchi et al., 2020*). In this article, **Morgan fingerprints** with a diameter of 4 and consisting of 512 bits are derived from RDKit (*Landrum, 2013*). The resulting vector is embedded with a linear layer to get a dense drug representation. Embedding drugs using such structural features overcomes the aforementioned drawbacks with one-hot embeddings.

Similarly, the identity of a drug can be communicated via the textual representation known as SMILES strings (*Weininger, 1988*). Here, an adaptation of SMILES for machine and deep learning applications is used, called DeepSMILES (*O'Boyle and Dalke, 2018*). All the different letters in the alphabet are assigned an index in a one-hot vector. Hence, every molecule can be encoded to a matrix $S_j \in \mathbb{R}^{v \times l}$, with $v$ the vocabulary size of the DeepSMILES alphabet and $l$ the string length of the molecule. This representation can be processed to a vector embedding using any neural network type that is appropriate for variable-length sequences.

A **1D CNN** detects and composes local patterns in the DeepSMILES string to a final drug embedding. Every input channel corresponds to a specific letter in the SMILES alphabet. The convolutional network used here consists of a position-wise linear layer to embed the channels to 64 dimensions, four convolutional blocks placed in sequence, followed by a global max-pooling operation across the length axis and a final linear layer to return a vector $t_j \in \mathbb{R}^{64}$. The global max-pooling layer allows the same network to be used for variable-length inputs. Each convolutional block consists of a structure similar to the one found in transformers (*Vaswani et al., 2017*). A first layer normalization is followed by a (padded) convolutional layer with a kernel size of 5, a first residual connection is wrapped around these two operations. After this, a position-wise feedforward makes up the second half of the convolutional block. The position-wise feedforward consists of a layer normalization, after which a GeLU-based gated linear unit identical to the one introduced by *Shazeer, 2020* is employed: $z = Dropout_{0.2}((GeLU(xW) \odot xV))W_2$. First, the input is sent to two position-wise linear layers via $W$ and $V$, each of them exploding the hidden dimensions of the input by a factor of 4. By sending the result of the first linear layer to a GeLU activation and multiplying element-wise with the result of the second linear layer, a gated linear unit structure is obtained. The output of this gated linear unit is sent to a dropout layer with rate 0.2 and then returned to the original dimension size via a final linear layer $W_2$. Around this second LayerNorm and feedforward structure, a residual connection is again wrapped. All residual blocks have an input and output hidden dimension of 64. *Appendix 2—figure 1* shows the structure of this convolutional block. Its design adopts the current state-of-the-art practices in transformers, which are increasingly being used in convolutional networks (*Liu et al., 2022*).

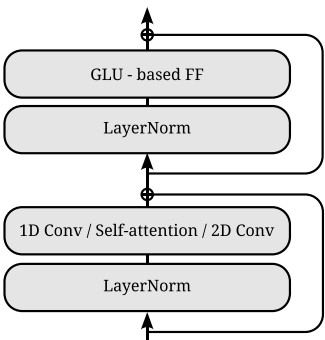

**Appendix 2—figure 1.** Structure used for the residual blocks, used in the 1D CNN, 2D CNN, and transformer. In the case of convolutions, the output is zero padded so as to produce the same output dimensions as in the input.

A **transformer** can be used to learn and compose signals in the DeepSMILES strings that occur sequence-wide, as opposed to the local pattern detection with a 1D CNN. The DeepSMILES strings are similarly embedded to 64 dimensions per character. After this, sinusoidal positional encodings (**Vaswani et al., 2017**) are added, and a CLS token embedding is prepended to the sequence. Four transformer blocks are employed, each with 64 as hidden dimension. The structure of the blocks are identical as with the 1D CNN (**Figure 1**), but using scaled dot-product self-attention instead of 1D convolutions. The self-attention operation uses eight heads. The output at the CLS token is used as a 'summary' of the content in the sequence (as opposed to the global max-pooling with the CNN). A final linear layer on the output of the CLS token returns the drug embedding $t_j \in \mathbb{R}^{64}$.

A **recurrent neural network** (RNN) is used to process the DeepSMILES strings sequentially. The RNN used here consists of a bidirectional GRU with 64 hidden dimensions (**Cho et al., 2014**). The two final hidden states of the GRU are used as 'summaries' of the content in the sequence. These two final states are averaged (element-wise) and sent to a final linear layer returning $t_j \in \mathbb{R}^{64}$.

All three aforementioned neural network structures work on variable-length (Deep)SMILES strings. With mini-batches, input drugs are (zero) padded so that everything fits into a tensor $\mathbf{S} \in \mathbb{R}^{b \times v \times l}$, with $b$ the batch size, $v$ the vocabulary size, and $l$ the longest length of a drug in the batch. The three aforementioned neural nets are adapted so that no information can flow from masked tokens to actual drug tokens (through masking after convolutions or in the attention matrices).

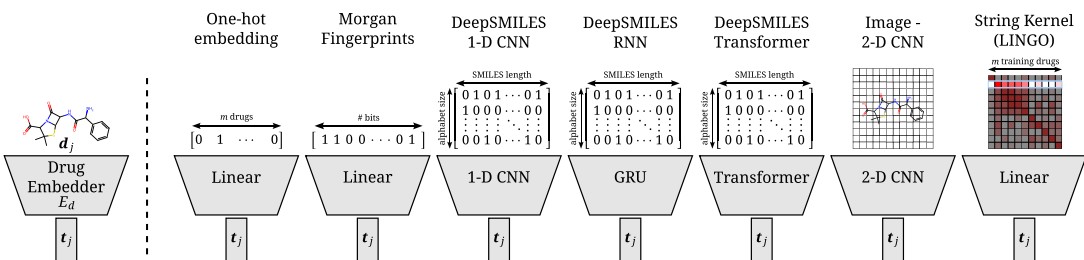

**Appendix 2—figure 2.** Overview of all different drug embedders tested in this work. One-hot embeddings are the only technique not incorporating prior knowledge of the structure of the compound. Hence, they are the only technique incapable of directly transferring to new compounds. Morgan fingerprints produce a bit-vector containing information on the presence of certain substructures. DeepSMILES strings are encoded and processed with a 1D CNN, GRU, or transformer. Drawings of molecules are processed with a 2D CNN. A string kernel on SMILES strings produces a numerical vector for every drug (taken as the row in the resulting Gram matrix).

As (Deep)SMILES are a 1D representation of a 3D molecular structure, a more detailed view of the drug may be obtained by permitting an extra dimension into its input representation. Drawings of drugs achieve this 2D view of the molecule. Here, 128 × 128 drawings of drugs are obtained through RDKit. The RGB values are inverted so as to make the parts of the image containing molecule 'activated'. Also, the RGB values are scaled to the range of 0–1 by dividing by 255.

A **2D CNN** processes the images to a drug embedding. The CNN consists of an input convolutional layer with kernel size and stride of 2. The input layer takes the three input channels and returns 32 hidden dimensions. Afterward, two convolutional blocks of the same structure as with the transformer

and 1D CNN are placed in tandem (*Appendix 2—figure 1*). The 2D convolutional operation used in the convolutional operation has a kernel size of 5. Hereafter, a global max-pooling operation across the height and width of the image is performed, followed by a final linear layer producing the drug embedding $t_j \in \mathbb{R}^{64}$.

A final way to obtain a numerical representation of drugs tested here is through similarity matrices. A **string kernel** is used to create a Gram matrix of all drugs in the training set. The input representation of a drug is then simply a row in said Gram matrix. This approach is generalizable to unseen drugs at inference time as obtaining a representation for them involves running the kernel function of the new compound to all training drugs. In this work, the LINGO string kernel (using 4-mers) is used (*Vidal et al., 2005*) as this kernel performed well in a recent benchmark (*Öztürk et al., 2016*). Note that here SMILES strings are used instead of DeepSMILES (as with the 1D CNN, RNN, and transformer). A linear layer produces the final drug embedding $t_j \in \mathbb{R}^{64}$ from a row in the Gram matrix.

A visual overview of all seven drug embedders in shown in *Appendix 2—figure 2*.

## Hyperparameter tuning
### Dual-branch models
Due to the complexity of tuning two branches and the size of the dataset, tuning is mostly done in an ad hoc fashion, relying on knowledge of current best practices in deep learning. Only some hyperparameters of interest are tuned on the validation set. Here, we present validation model results of those experiments. All results presented here concern models that are trained with a medium-sized spectrum embedder, with hyperparameters otherwise as described in Appendix 2. All numbers indicate an average over five runs, similarly choosing the best average out of four tested learning rates (1e-5, 5e-5, 1e-4, 5e-4, 1e-3, 5e-3).

*Appendix 2—figure 3A* shows validation set performances for a grid of different kernel sizes and hidden dimensionalities for the SMILES 1-D CNN. The best-performing hidden dimensionality (64) is copied to the (Deep)SMILES Transformer and GRU without further tuning. In *Appendix 2—figure 3B*, a similar grid is shown for the Image 2-D CNN, where it is found that a smaller hidden size is favored. *Appendix 2—figure 3C* shows the performance for using different molecular string representations as input to the 1-D CNN model: SMILES, DeepSMILES (*O'Boyle and Dalke, 2018*), and SELFIES (*Krenn et al., 2020*). While all techniques perform competitively, DeepSMILES strings outperform the other two by a small margin. Similarly, DeepSMILES are thus selected as input representations for the Transformer and GRU, without further tuning. *Appendix 2—figure 3D* shows how sinusoidal positional encodings outperform learned positional encodings (as in *Devlin et al., 2018*). It is found that a bidirectional GRU considerably outperforms a unidirectional one (*Appendix 2—figure 3E*). Finally, the number of bits in the Morgan fingerprint encoding is also tuned (*Appendix 2—figure 3F*). It is seen that including lower than 512 bits degenerates performance, but including more than 512 introduces instabilities in model training, as the model becomes prone to overfitting the drug branch.

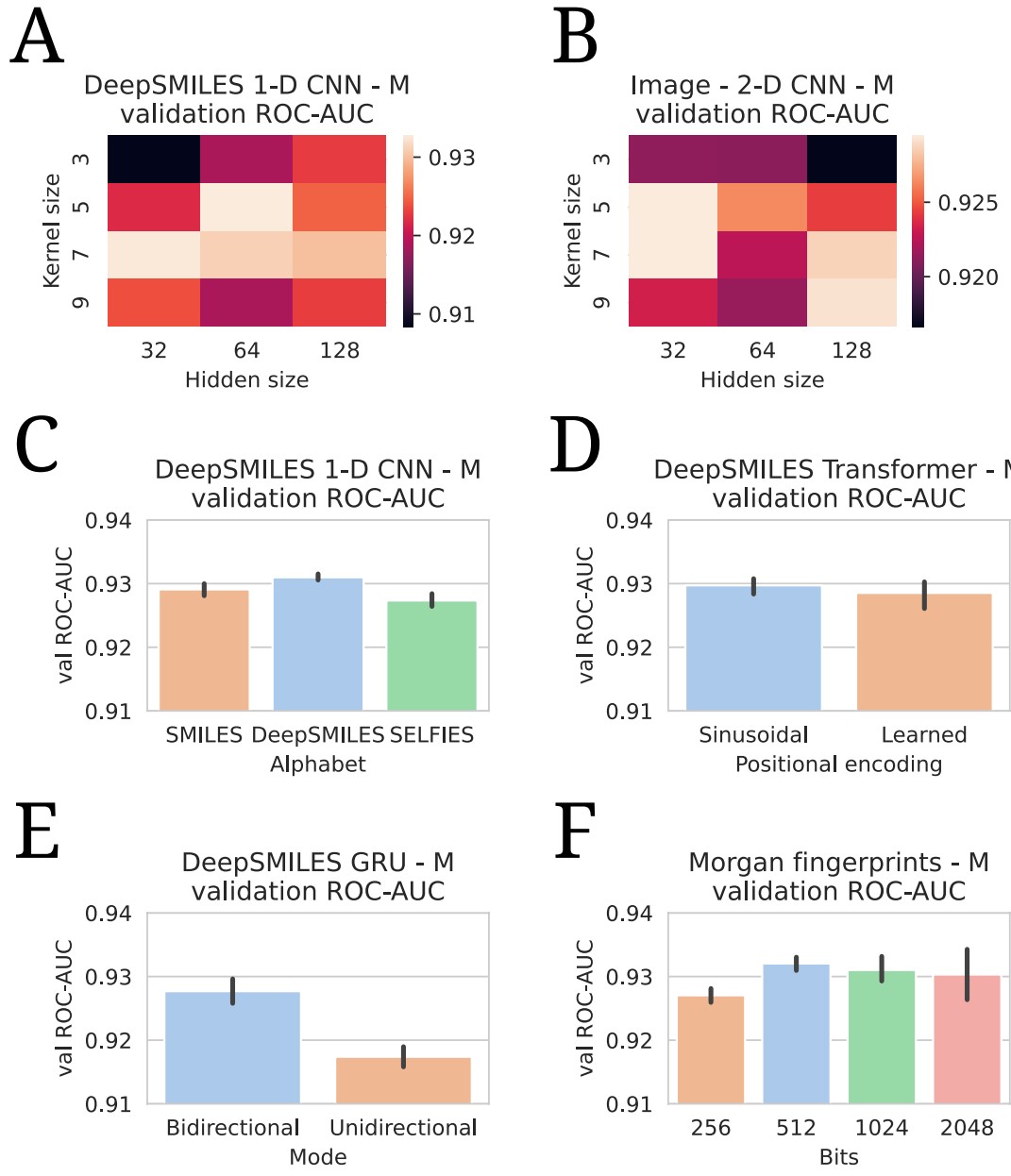

**Appendix 2—figure 3.** All hyperparameter tuning experiments. All evaluations are listed in terms of validation area under the receiver operating characteristic curve (ROC-AUC). All numbers are averages of five model runs, with errorbars showing standard deviations. In every experiment, the highest average is chosen to use in the final models. (**A**) Tuning of kernel and hidden size in a DeepSMILES CNN. (**B**) Tuning of kernel and hidden size in an Image CNN. (**C**) Tuning of alphabet in a DeepSMILES CNN. (**D**) Tuning of positional encodings in a DeepSMILES Transformer. (**E**) Tuning of directionality in a DeepSMILES GRU. (**F**) Tuning of number of bits in a Morgen Fingerprint-based drug embedder.

## Specialist baselines

All baselines are trained using the same data splits as used with the dual-branch model. In essence: all DRIAMS-A spectra before the year 2018 are in the training set. The remaining spectra from 2018 are evenly divided among validation and test (with which belonging to which corresponding with the splits used for the dual-branch experiments). The same preprocessed 6000-dimensional spectrum representations are used as input.

Logistic regression baselines are trained with the L-BFGS solver for a maximum of 500 training iterations. For every species–drug combination, a grid search is performed on various hyperparameters, selecting the best based on validation ROC-AUC. The hyperparameters that are tuned are the scaling method on the features (either none, or standard scaling), and the L2 regularization strength ($C \in \{10^{-3}, 10^{-2}, ..., 10^{2}, 10^{3}\}$).

For XGBoost, default parameters are used apart from those tuned. For every species–drug combination, a grid is run, testing different numbers of trees (`n_estimators` $\in \{25, 50, 100, 200\}$) and learning rate (`learning_rate` $\in \{10^{-3}, 10^{-2}, 10^{-1}, 10^{0}\}$).

For the MLP baselines, the same hyperparameters are used as for the spectrum branch. Briefly recapitulated: between every two fully connected layers, a series of operations consisting of (1) a GeLU activation, (2) a dropout rate of 0.2, and (3) layer normalization is applied. The sizes of the models are as in *Table 1*, but then ending in 1 node instead of 64. For every species–drug combination, models are trained using the cross-entropy loss and the Adam optimizer for a maximum of 250 epochs. A batch size of 128 is employed. A linear learning rate warm-up is applied over the first 250 steps. Early stopping based on validation ROC-AUC is applied with a patience of 10 epochs. The model with the best validation ROC-AUC during training is kept as final model. the best model out of four different learning rates (learning_rate $\in \{1e\text{-}5, 5e\text{-}5, 1e\text{-}4, 5e\text{-}4\}$) is chosen based on their validation ROC-AUC.

# Appendix 3

## Tables and figures supporting the results section

**Appendix 3—table 1.** Full table of test results.

The listed averages and standard deviations are calculated over five independent runs of the same model. The best models for every metric per drug embedder are underlined. The overall best model for every metric is in bold face.

| Drug embedder | Spectrum embedder | ROC-AUC | Prec@1(-) | Macro ROC-AUC |
|---|---|---|---|---|
| Morgan fingerprints | S | 0.9341 ± 0.0014 | 0.9917 ± 0.0009 | **0.8158 ± 0.0070** |
|  | M | **0.9345 ± 0.0014** | **0.9922 ± 0.0009** | 0.8078 ± 0.0081 |
|  | L | 0.9341 ± 0.0007 | 0.9920 ± 0.0010 | 0.8070 ± 0.0128 |
|  | XL | 0.9322 ± 0.0017 | 0.9920 ± 0.0012 | 0.7904 ± 0.0155 |
| One-hot embedding | S | 0.9326 ± 0.0017 | 0.9899 ± 0.0010 | 0.7984 ± 0.0086 |
|  | M | 0.9337 ± 0.0014 | 0.9910 ± 0.0016 | 0.7920 ± 0.0175 |
|  | L | 0.9338 ± 0.0018 | 0.9882 ± 0.0010 | 0.8011 ± 0.0116 |
|  | XL | 0.9327 ± 0.0011 | 0.9890 ± 0.0026 | 0.7932 ± 0.0201 |
| DeepSMILES 1-D CNN | S | 0.9303 ± 0.0012 | 0.9864 ± 0.0016 | 0.7949 ± 0.0185 |
|  | M | 0.9336 ± 0.0011 | 0.9903 ± 0.0008 | 0.8009 ± 0.0044 |
|  | L | 0.9337 ± 0.0015 | 0.9890 ± 0.0014 | 0.7940 ± 0.0052 |
|  | XL | 0.9317 ± 0.0012 | 0.9898 ± 0.0020 | 0.7960 ± 0.0155 |
| String Kernel (LINGO) | S | 0.9327 ± 0.0022 | 0.9913 ± 0.0012 | 0.7972 ± 0.0087 |
|  | M | 0.9332 ± 0.0017 | 0.9916 ± 0.0008 | 0.7919 ± 0.0051 |
|  | L | 0.9317 ± 0.0017 | 0.9909 ± 0.0013 | 0.7859 ± 0.0136 |
|  | XL | 0.9303 ± 0.0021 | 0.9893 ± 0.0025 | 0.7935 ± 0.0135 |
| Image – 2-D CNN | S | 0.9310 ± 0.0016 | 0.9888 ± 0.0025 | 0.7820 ± 0.0101 |
|  | M | 0.9317 ± 0.0008 | 0.9885 ± 0.0019 | 0.7866 ± 0.0084 |
|  | L | 0.9332 ± 0.0010 | 0.9901 ± 0.0016 | 0.7758 ± 0.0070 |
|  | XL | 0.9309 ± 0.0012 | 0.9900 ± 0.0013 | 0.7711 ± 0.0109 |
| DeepSMILES Transformer | S | 0.9306 ± 0.0021 | 0.9900 ± 0.0022 | 0.7862 ± 0.0124 |
|  | M | 0.9325 ± 0.0012 | 0.9891 ± 0.0014 | 0.7925 ± 0.0075 |
|  | L | 0.9308 ± 0.0014 | 0.9885 ± 0.0027 | 0.7902 ± 0.0072 |
|  | XL | 0.9311 ± 0.0014 | 0.9895 ± 0.0009 | 0.7791 ± 0.0075 |
| DeepSMILES RNN | S | 0.9291 ± 0.0015 | 0.9872 ± 0.0032 | 0.7881 ± 0.0053 |
|  | M | 0.9293 ± 0.0028 | 0.9863 ± 0.0008 | 0.7793 ± 0.0116 |
|  | L | 0.9266 ± 0.0012 | 0.9868 ± 0.0019 | 0.7684 ± 0.0058 |
|  | XL | 0.9278 ± 0.0029 | 0.9879 ± 0.0027 | 0.7689 ± 0.0113 |

ROC-AUC area under the receiver operating characteristic curve.

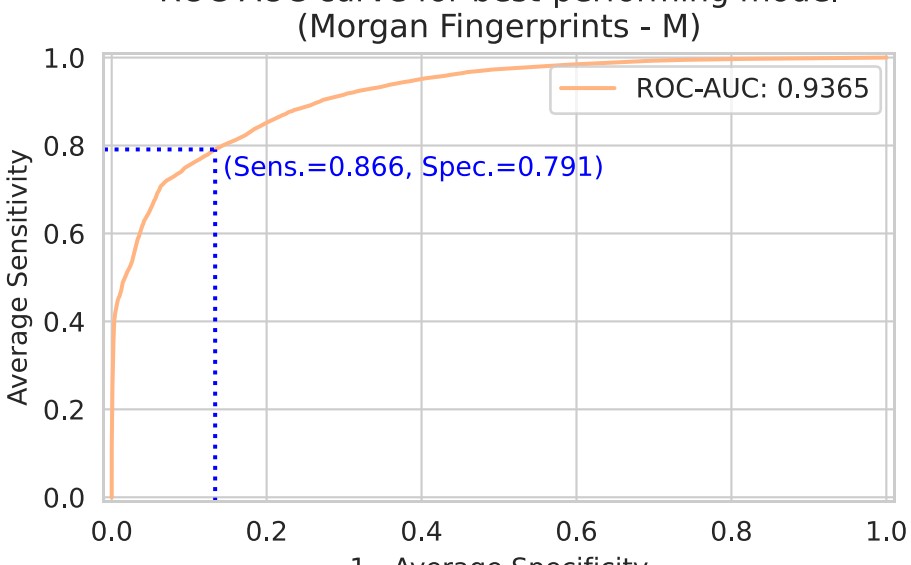

**Appendix 3—figure 1.** Spectrum-macro receiver operating characteristic (ROC) curve for best-performing model (Morgan fingerprints drug embedder, medium-sized spectrum embedder). The y-axis shows the average sensitivity (across patients), while the x-axis shows one minus the average specificity. Note that this ROC curve is not a traditional ROC curve constructed from one single label set and one corresponding prediction set. Rather, it is constructed from spectrum-macro metrics as follows: for any possible threshold value, binarize all predictions. Then, for every spectrum/ patient independently, compute the sensitivity and specificity for the subset of labels corresponding to that spectrum/ patient. Finally, those sensitivities and specificities are averaged across patients to obtain one point on above ROC curve. In blue, the optimal sensitivity and specificity (according to the Youden index) is indicated (*Youden, 1950*).

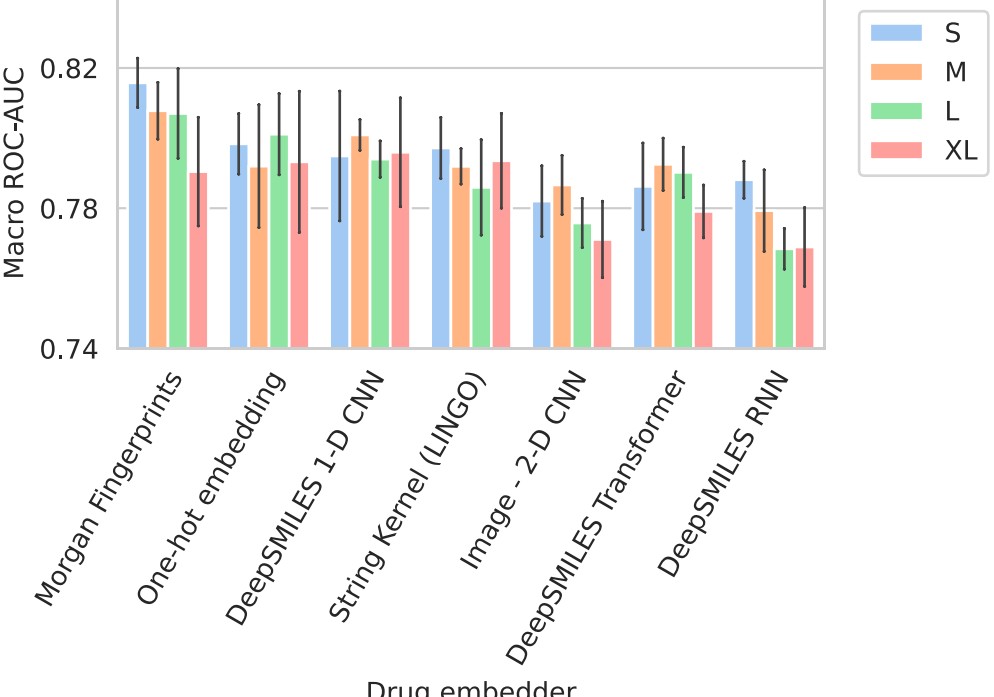

**Appendix 3—figure 2.** Barplots showing test performance results for all trained models. Colors represent the different spectrum embedder model sizes. Performance is shown in terms of macro area under the receiver operating characteristic curve (ROC-AUC) (computed per drug and averaged). Errorbars represent the standard deviation over five random seeds.

**Appendix 3—table 2.** Test area under the receiver operating characteristic curve (ROC-AUC) performance per species.

Reported figures are averages across the five different medium-sized Morgan fingerprint-based recommenders.

| Species | ROC-AUC |
|---|---|
| *Staphylococcus aureus* | 0.9578 |
| *Staphylococcus epidermidis* | 0.9478 |
| *Escherichia coli* | 0.9184 |
| *Klebsiella pneumoniae* | 0.9643 |
| *Pseudomonas aeruginosa* | 0.7614 |
| *Enterobacter cloacae* | 0.9831 |
| *Proteus mirabilis* | 0.9727 |
| *Staphylococcus hominis* | 0.9594 |
| *Serratia marcescens* | 0.9848 |
| *Staphylococcus capitis* | 0.9425 |
| *Enterococcus faecium* | 0.9914 |
| *Klebsiella oxytoca* | 0.9861 |
| *Klebsiella variicola* | 0.9824 |
| *Citrobacter koseri* | 0.9970 |
| *Enterococcus faecalis* | 0.9594 |
| *Staphylococcus lugdunensis* | 0.9705 |
| *Citrobacter freundii* | 0.9622 |
| *Morganella morganii* | 0.9931 |
| *Proteus vulgaris* | 0.9828 |
| *Staphylococcus haemolyticus* | 0.9751 |
| *Candida albicans* | 0.7446 |
| *Streptococcus pneumoniae* | 0.9059 |
| *Stenotrophomonas maltophilia* | 1.0000 |
| *Campylobacter jejuni* | 1.0000 |
| *Haemophilus influenzae* | 1.0000 |

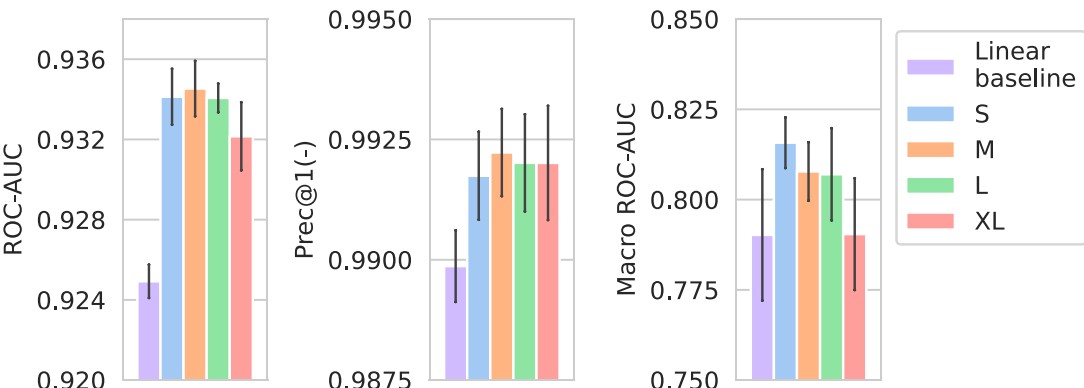

**Appendix 3—figure 3.** Performance of models compared against a linear spectrum embedder baseline. The comparison is only shown for the best-performing drug embedder (Morgan fingerprints). Errorbars represent the standard deviation over five random seeds.

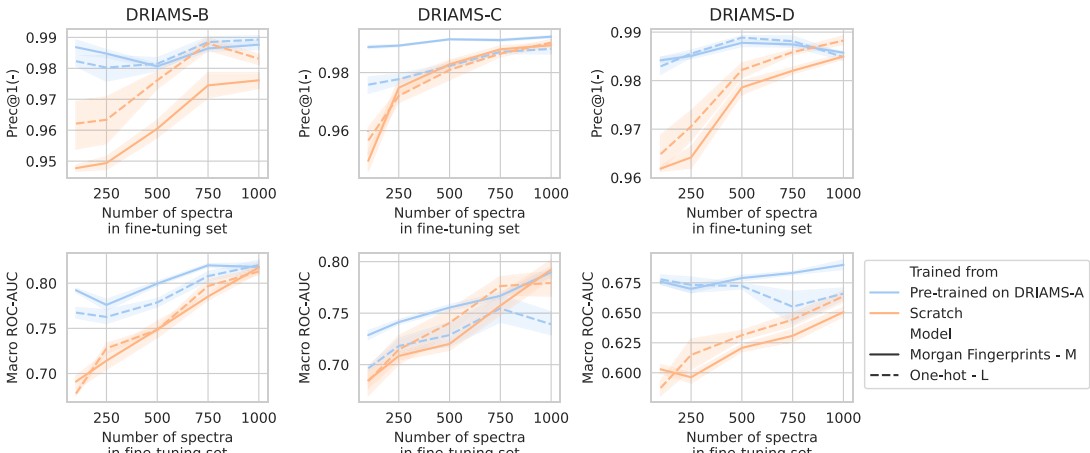

**Appendix 3—figure 4.** Transfer learning of DRIAMS-A models to other hospitals. Errorbands show the standard deviation over five runs.

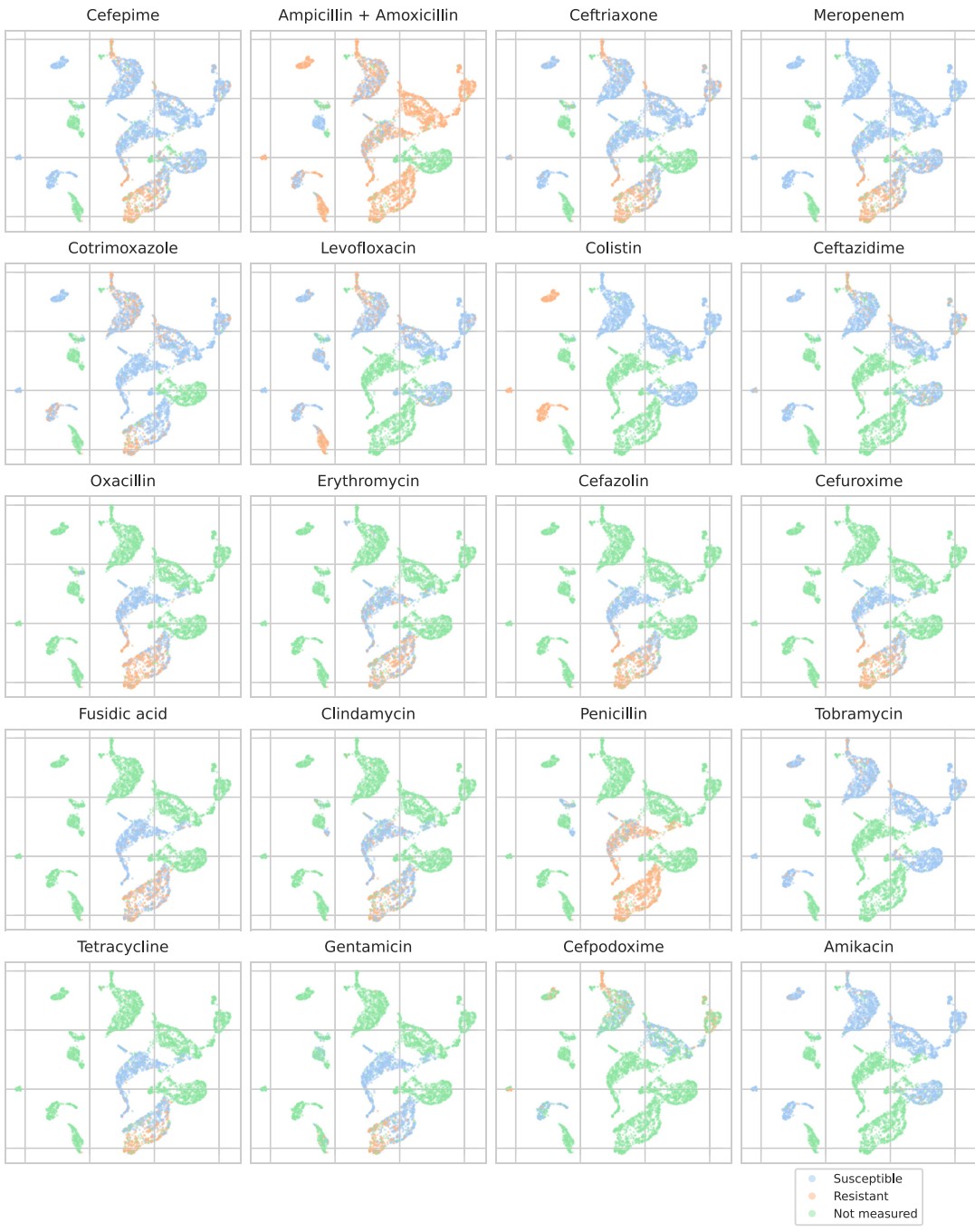

**Appendix 3—figure 5.** UMAP scatterplots of test set matrix-assisted laser desorption/ionization time-of-flight (MALDI-TOF) spectra embeddings $x_i$. Embeddings from a 'general' (trained on all spectra across species) recommender are shown. Only embeddings belonging to the 25 most occurring species in the test set are shown. Spectra are colored according to its antimicrobial resistance (AMR) status to a certain drug. The 20 displayed drugs were selected based on a ranking of the product of the number of positive and negative labels $\sum_{i=1}^{n}[y_{ij} = 0] \cdot \sum_{i=1}^{n}[y_{ij} = 1]$. In this way, the drugs that have a lot of observed labels, both positives and negatives, are displayed. The drugs here are ranked 5–24 (the first four are shown in **Figure 4**). In order to map the clusters back to species, readers are referred back to **Figure 4**.

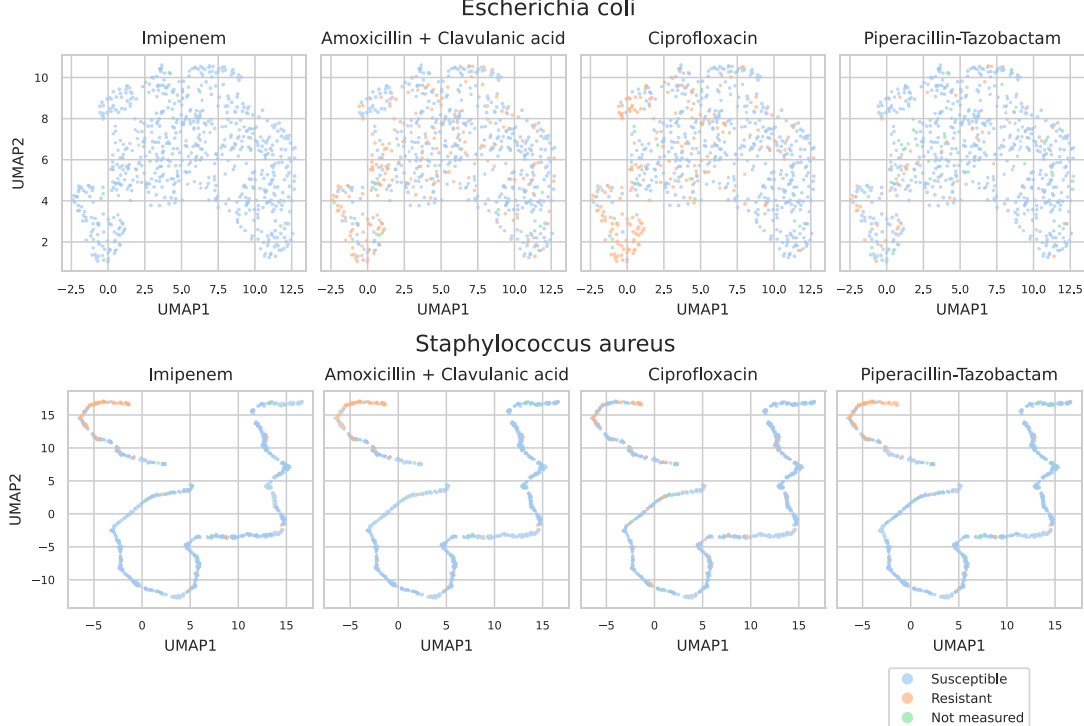

**Appendix 3—figure 6.** UMAP scatterplots of test set matrix-assisted laser desorption/ionization time-of-flight (MALDI-TOF) spectra embeddings $x_i$. Embeddings from two 'species-specific' recommenders are shown. Spectra are colored according to its antimicrobial resistance (AMR) status to a certain drug.

