## [Editor Report · eLife Assessment]

This **valuable** study presents a machine learning model to recommend effective antimicrobial drugs from patients' samples analyzed with mass spectrometry. The evidence supporting the claims of the authors is **convincing**. This work will be of interest to computational biologists, microbiologists, and clinicians.

---

## [Referee Report · Joint public reviews]

De Waele et al. framed the mass-spectrum-based prediction of antimicrobial resistance (AMR) prediction as a drug recommendation task. Neural networks were trained on the recently available DRIAMS database of MALDI-TOF (matrix-assisted laser desorption/ionization time-of-flight) mass spectrometry data and their associated antibiotic susceptibility profiles (Weis et al. 2022). Weis et al. (2022) also introduced the benchmark models which take as the input a single species and are trained to predict resistance to a single drug. Instead here, a pair of drugs and spectrum are fed to two neural network models to predict a resistance probability. In this manner, knowledge from different drugs and species can be shared through the model parameters. Questions asked: What is the best way to encode the drugs? Does the dual neural network outperform the single spectrum-drug network?

The authors showed consistent performance of their strategy to predict antibiotic susceptibility for different spectrum and antibiotic representations (i.e., embedders). Remarkably, the authors showed how small datasets collected at one location can improve the performance of a model trained with limited data collected at a second location. The authors also showed that species-specific models (trained in multiple antibiotic resistance profiles) outperformed both the single recommender model and the individual species-antibiotic combination models.

Strengths:

• A single antimicrobial resistance recommender system could potentially facilitate the adoption of MALDI-TOF based antibiotic susceptibility profiling into clinical practices by reducing the number of models to be considered, and the efforts that may be required to periodically update them.

• The authors tested multiple combinations of embedders for the mass spectra and antibiotics while using different metrics to evaluate the performance of the resulting models. Models trained using different spectrum embedder-antibiotic embedder combinations had remarkably good performance for all tested metrics. The average ROC AUC scores for global and species-specific evaluations were above 0.8.

• Authors developed species-specific recommenders as an intermediate layer between the single recommender system and single species-antibiotic models. This intermediate approach achieved maximum performance (with one type of the species-specific recommender achieving a 0.9 ROC AUC), outlining the potential of this type of recommenders for frequent pathogens.

• Authors showed that data collected in one location can be leveraged to improve the performance of models generated using a smaller number of samples collected at a different location. This result may encourage researchers to optimize data integration to reduce the burden of data generation for institutions interested in testing this method.

Weaknesses:

• Authors do not offer information about the model features associated with resistance. While reviewers understand that it is difficult to map mass spectra to specific pathways or metabolites, mechanistic insights are much more important in the context of AMR than in the context of bacterial identification. For example, this information may offer additional antimicrobial targets. Thus, authors should at least identify mass spectra peaks highly associated with resistance profiles. Are those peaks consistent across species? This would be a key step towards a proteomic survey of mechanisms of AMR. See previous work on this topic (Hrabak et al. 2013, Torres-Sangiao et al. 2022).

References:

Hrabak et al. (2013). Clin Microbiol Rev 26. doi: 10.1128/CMR.00058-12.

Torres-Sangiao et al. (2022). Front Med 9. doi: 10.3389/fmed.2022.850374.

Weis et al. (2022). Nat Med 28. doi: 10.1038/s41591-021-01619-9.

---

## [Author Response]

The following is the authors’ response to the previous reviews.

Reviewer #1:Section 4.3 ("expert baseline model"): the authors need to explain how the probabilities defined as baselines were exactly used to predict individual patient susceptible profiles.

We have added a more detailed and mathematically formal explanation of the “simulated expert’s best guess” in Section 4.3.

This section now reads:

“More formally, considering all training spectra as *Strain*, all training labels corresponding to one drug *j* and species *t* are gathered: ysubset j,t={yij∣si∈Strain ∧species⁡(si)=t}. The "simulated expert's best guess" predicted probability for any spectrum *si* and drug *dj*, then, corresponds to, the fraction of positive labels in their corresponding training label set Ysubset j,t: ysubset j,t={yij∣si∈Strain ∧species⁡(si)=t}”

Authors should explain in more detail how a ROC curve is generated from a single spectrum (i.e., per patient) and then average across spectra. I have an idea of how it's done but I am not completely sure.

We have added a more detailed explanation in Section 3.2. It reads:

To compute the (per-patient average) ROC-AUC, for any spectrum/patient, all observed drug resistance labels and their corresponding predictions are gathered. Then, the patient-specific ROC-AUC is computed on that subset of labels and predictions. Finally, all ROC-AUCs per patient are averaged to a "spectrum-macro" ROC-AUC.

In addition, our description under Supplementary Figure 8 (showing the ROC curve) provides additional clarification:

Note that this ROC curve is not a traditional ROC curve constructed from one single label set and one corresponding prediction set. Rather, it is constructed from spectrum-macro metrics as follows: for any possible threshold value, binarize all predictions. Then, for every spectrum/patient independently, compute the sensitivity and specificity for the subset of labels corresponding to that spectrum/patient. Finally, those sensititivies and specificities are averaged across patients to obtain one point on above ROC curve.

Section 3.2 & reply # 1: can the authors compute and apply the Youden cutoff that gives max precision-sensitivity for each ROC curve? In that way the authors could report those values.

We have computed this cut-off on the curve shown in Supplementary Figure 8. The Figure now shows the sensitivity and specificity at the Youden cutoff in addition to the ROC. We have chosen only to report these values for this model as we did not want to inflate our manuscript with additional metrics (especially since the ROC-AUC already captures sensitivities and specificities). We do, however, see the value of adding this once, so that biologists have an indication of what kind of values to expect for these metrics.

Related to reply #5: assuming that different classifiers are trained in the same data, with the same number of replicates, could authors use the DeLong test compare ROC curves? If not, please explain why.

We thank the reviewer for bringing our attention to the DeLong’s test. It does indeed seem true that this test is appropriate for comparing two ROC-AUCs using the same ground truth values.

We have chosen not to use this test for one conceptual and one practical reason:

(1) Our point still stands that in machine learning one chooses the test set, and hence one can artificially increase statistical power by simply allocating a larger fraction of the data to test.

(2) DeLong’s test is defined for single AUCs (i.e. to compare two lists of predictions against one list of ground truths), but here we report the spectrum/patient-macro ROC-AUC. It is not clear how to adjust the test to macro-evaluated AUCs. One option may be to apply the test per patient ROC curve, and perform multiple testing correction, but then we are not comparing models, but models per patient. In addition, the number of labels/predictions per patient is prohibitively small for statistical power.

**Reviewer #2 (Recommendations For The Authors):**

After revision, all issues were been resolved.